# A New Proportionate Filtered-x RLS Algorithm for Active Noise Control System

**DOI:** 10.3390/s22124566

**Published:** 2022-06-17

**Authors:** Xiaobei Liang, Jinyong Yao, Lei Luo, Wenzhao Zhu, Weifang Zhang, Yanrong Wang

**Affiliations:** 1School of Energy and Power Engineering, Beihang University, Beijing 100191, China; liangxiaobei2016@163.com (X.L.); yrwang@buaa.edu.cn (Y.W.); 2School of Reliability and Systems Engineering, Beihang University, Beijing 100191, China; jinyongyao@buaa.edu.cn (J.Y.); 08590@buaa.edu.cn (W.Z.); 3Key Laboratory of Optoelectronic Technology and Systems, Education Ministry of China, Chongqing University, Chongqing 400044, China; 4School of Electrical Engineering, Yanshan University, Qinhuangdao 066004, China; wenzhaozhu@ysu.edu.cn

**Keywords:** active noise control, FxRLS, tracking performance, momentum technique, convergence condition

## Abstract

The filtered-x recursive least square (FxRLS) algorithm is widely used in the active noise control system and has achieved great success in some complex de-noising environments, such as the cabin in vehicles and aircraft. However, its performance is sensitive to some user-defined parameters such as the forgetting factor and initial gain. Once these parameters are not selected properly, the de-noising effect of FxRLS will deteriorate. Moreover, the tracking performance of FxRLS for mutation is still restricted to a certain extent. To solve the above problems, this paper proposes a new proportional FxRLS (PFxRLS) algorithm. The forgetting factor and initial gain sensitivity are successfully reduced without introducing new turning parameters. The de-noising level and tracking performance have also been improved. Moreover, the momentum technique is introduced in PFxRLS to further improve its robustness and de-noising level. To ensure stability, its convergence condition is also discussed in this paper. The effectiveness of the proposed algorithms is illustrated by simulations and experiments with different user-defined parameters and time-varying noise environments.

## 1. Introduction

The active noise control (ANC) system is an electro-acoustic device, which can generate an anti-noise with the same amplitude, frequency and opposite phase of the undesired noise based on the principle of destructive interference. The core of the ANC system is the control algorithm. In order to improve the noise reduction performance of the ANC system, many corresponding adaptive algorithms have been developed and achieved great success. Kuo and Liu discussed the effectiveness of filtered-x least mean-squares (FxLMS) to cancel both the outside and inside noises of infant incubators in a real NICU environment, and they obtained an average reduction of 13 dB and 17 dB [1,2]. Mylonas developed a simple, robust and computationally efficient virtual sensing FxLMS algorithm and tested it in a cabin mock-up [3]. Zhang proposed a normalized frequency-domain block FxLMS (NFB-FxLMS) to reduce the interior noise of vehicles and showed that the higher the vehicle speed, the more obvious the advantages of the NFB-FxLMS [4]. Jia proposed a modified hybrid ANC algorithm combining the vibrational and acoustical signals for the reduction of road noise inside a vehicle cabin and achieved a 3.7 dBA overall noise reduction [5]. Reddy presented the application of filtered-x recursive least-squares (FxRLS), filtered-x affine projection algorithm (FxAPA) and FxLMS on an fMRI bore test-bed [6] and developed a new hybrid adaptive algorithm to balance the convergence and steady-state error based on the combination of FxRLS and FxNLMS [7]. Wu presented a hybrid active and passive noise control in the ventilation duct with an internally placed microphones module, showing that ANC can obtain more than 10 dB noise attenuation from 70 to 200 Hz when the wind speed increases from 5 to 10 m/s [8]. Sun discussed a spatial ANC system utilizing multiple circular microphones and loudspeaker arrays for a 3D space and achieved good noise-reduction performance [9]. Luo proposed a Volterra recursive even mirror Fourier nonlinear filter for the nonlinear ANC system with acoustic feedback, showing that it can achieve global noise control in closed space [10]. Kim proposed a new algorithm to enhance the convergence rate and noise reduction efficiency of ANC headsets with fixed-point DSP, reducing the broadband noise by 24 dB [11]. We notice that the FxLMS and FxRLS-based adaptive algorithms are the two of the most widely used in ANC systems. However, it is well known that the main limitations of LMS-based algorithms are their lower convergence rate than RLS-based algorithms in general, and they have a larger steady-state mean square error as compared to RLS-based algorithms [7,12,13]. Conversely, RLS-based algorithms need much more computational load. With the development of high-speed processing devices, the impact of this computational load becomes negligible. Thus, the FxRLS algorithm has gradually become a focus of attention.

In practical application, the FxRLS algorithm involves many user-defined parameters, which are closely related to its performance [14,15]. The forgetting factor and the initial gain are two such parameters. The forgetting factor is used to balance the importance of recent data and older data. A constant forgetting factor yields a fixed penalty. The forgetting factor usually chooses from 0.9 to 0.99, and a large value yields a slow convergence rate and low steady-state error. Whereas for a smaller forgetting factor, FxRLS holds a faster convergence performance and relatively higher steady-state error. The initial gain is the initial value for the gain matrix. As was pointed out by Moustakides [16], there is a variable performance of RLS as a function of the initialization of initial gain. In some cases, RLS can exhibit a significantly faster convergence when initialized with a relatively small positive definite value than when initialized with a large one. Generally, for the cases of high and medium signal-to-noise ratio (SNR), a relatively small initial gain can obtain a better performance. Whereas for low SNR, a relatively large value must be selected for achieving the best results. In [17], it is also observed that the value of the initializing constant initial gain also plays an important role in the convergence performance. Consequently, choosing suitable values for these pre-set parameters is typically a trial and error process [18]. In fact, however, the SNR is unknown, and the noise environment may be non-stationary, such that these pre-set constant parameters are not enough to meet the actual demand. Moreover, it is very difficult or rather impossible for FxRLS to be re-initialized during runtime application.

To solve the influence of pre-set constant parameters, many variable-parameter techniques have been proposed, where the variable forgetting factor is the most representative one among them. In particular, Kohli presented a numeric variable forgetting factor RLS algorithm for first-order and second-order time-varying Volterra systems in 2013 [19]. Chu proposed a novel diffusion variable forgetting factor RLS algorithm based on a local polynomial modeling in 2017 [20]. Chan reported a new variable forgetting factor based on total noise variance and variance of the impulse response vector in 2020 [21]. Bernstein derived a variable forgetting factor by minimizing a quadratic cost function in 2020 [22]. The reported methods above improved the influence of constant forgetting factor, but they are relatively complex and introduce more user-defined parameters. Once these new parameters are chosen inappropriately, their performance will deteriorate. Therefore, introducing new user-defined parameters to reduce the effect of the forgetting factor on the performance of FxRLS is not a better solution. This method will produce a new trial-and-error process. As for another user-defined parameter, initial gain, the reported works mainly use an smaller constant value, ignoring its impact on the performance of FxRLS. In fact, the performance of FxRLS is very sensitive to initial gain, especially in a time-varying noise environment. Once the value is not correct, their convergence will be delayed. However, so far, few studies are designed to reduce the effect of initial gain on the performance of RLS-based algorithms. The variable parameter technique for initial gain is a solution. For the ANC system, however, it is rather difficult to employ multiple variable parameter methods. Furthermore, these variable parameter methods based on introducing new user-defined parameters will also produce a new parameter selection problem. Therefore, under the condition without introducing a new parameter, how to alleviate the trial and error problems of forgetting factor and initial gain at the same time is still a tough question.

In order to overcome the problems above, this paper proposes a new proportionate FxRLS (PFxRLS) structure. In PFxELS, the forgetting factor is associated with the difference between the posterior error and the prior error. By gradient calculation, a proportional factor related to the forgetting factor is derived in the controller update. The PFxRLS can mitigate the effect of choosing the forgetting factor and initial gain, and it can keep good de-noising performance in a wide parameter range. More importantly, compared to other improved FxRLS algorithms, the PFxRLS does not introduce new user-defined parameters. Furthermore, the PFxRLS also improve the de-noising level to some extent compared to standard FxRLS. Based on the PFxRLS, the momentum technique is introduced to further improve its performance. The numerical examples and experiment show momentum PFxRLS (MPFxRLS) can achieve a lower steady-state error under the condition of the same convergence speed.

The rest of the paper is organized as follows. Section 2 presents the standard FxRLS and the proposed algorithms including the PFxRLS and MPFxRLS. The convergence analysis and complexity analysis are also shown in Section 2. Section 3 provides the simulation studies. Section 4 presents the experiment results. Conclusions are given in Section 5.

## 2. Proposed Scheme

### 2.1. Preliminaries

The ANC system using the FxRLS algorithm is shown in Figure 1, where x(n) represents the reference signal, y(n) represents the secondary output, p(n) represents the primary noise, $s=[s_{0},\;s_{1},\;\cdots ,\;s_{M-1}]$ represents the secondary path, and $\hat{s}$ represents the estimated secondary path, generally $\hat{s}=s$. $w\left(n\right)={\left[w_{0}\left(n\right),\;w_{1}\left(n\right),\;\cdots ,\;w_{N-1}\left(n\right)\right]}^{T}$ represents the adaptive controller, $\hat{x}\left(n\right)$ given by
(1)$$\begin{array}{c} \hat{x}\left(n\right)=\sum_{m=0}^{M-1}s_{m}x\left(n-m\right), \end{array}$$
represents the filtered reference signal, e(n) represents the error signal, expressed as
(2)$$\begin{array}{c} e\left(n\right)=p\left(n\right)-{\hat{x}}^{T}\left(n\right)w\left(n-1\right), \end{array}$$
which is a prior error, and
(3)$$\begin{array}{c} \hat{x}\left(n\right)={\left[\hat{x}\left(n\right),\;\hat{x}\left(n-1\right),\;\cdots ,\;\hat{x}\left(n-N+1\right)\right]}^{T}. \end{array}$$

### 2.2. PFxRLS Algorithm

Define a new cost function
(4)$$\begin{array}{cc} J_{n}\left(w\right) & \overset{▵}{=}\sum_{i=0}^{n}\lambda^{n-i}\varepsilon^{2}\left(i\right)+(1-{\lambda )(e(n)-\varepsilon \left(n\right))}^{2}, \end{array}$$
where λ represents the forgetting factor, ε(n) is the posterior error, defined as
(5)$$\begin{array}{c} \varepsilon \left(n\right)\overset{▵}{=}p\left(n\right)-{\hat{x}}^{T}\left(n\right)w\left(n\right). \end{array}$$

Define
(6)$$\begin{array}{c} A\left(n\right)\overset{▵}{=}\sum_{i=0}^{n}\lambda^{n-i}\hat{x}\left(i\right){\hat{x}}^{T}\left(i\right), \end{array}$$
(7)$$\begin{array}{c} B\left(n\right)\overset{▵}{=}\sum_{i=0}^{n}\lambda^{n-i}\hat{x}\left(i\right)p\left(i\right), \end{array}$$
(8)$$\begin{array}{c} C\left(n\right)\overset{▵}{=}\sum_{i=0}^{n}\lambda^{n-i}p^{2}\left(i\right), \end{array}$$
then (Equation 4) becomes
(9)$$\begin{array}{cc} J_{n}\left(w\right) & =w^{T}A\left(n\right)w+\left(1-\lambda \right)\Delta w^{T}x\left(n\right)x^{T}\left(n\right)\Delta w-2B^{T}\left(n\right)w+C\left(n\right). \end{array}$$
where
(10)$$\begin{array}{c} \Delta w\overset{▵}{=}w\left(n\right)-w\left(n-1\right). \end{array}$$

Since (Equation 4) is quadratic and strictly convex, its unique global minimizer is the only local minimizer. It follows from $\frac{\partial J_{n}\left(w\right)}{\partial w}=0$ that the local minimizer is
(11)$$\begin{array}{c} w\left(n\right)=\frac{\left(1-\lambda \right)x^{T}\left(n\right)P\left(n\right)x\left(n\right)w\left(n-1\right)+P\left(n\right)B\left(n\right)}{1+\left(1-\lambda \right)x^{T}\left(n\right)P\left(n\right)x\left(n\right)}, \end{array}$$
where
(12)$$\begin{array}{c} P\left(n\right)\overset{▵}{=}A^{-1}\left(n\right), \end{array}$$
and P(0)=δI, δ>0 is initial gain. It follows from (Equation 6) that
(13)$$\begin{array}{c} A\left(n\right)=\lambda A\left(n-1\right)+\hat{x}\left(n\right){\hat{x}}^{T}\left(n\right), \end{array}$$
then,
(14)$$\begin{array}{c} P^{-1}\left(n\right)=\lambda P^{-1}\left(n-1\right)+\hat{x}\left(n\right){\hat{x}}^{T}\left(n\right). \end{array}$$

Using matrix inversion lemma, we have
(15)$$\begin{array}{cc} P(n)\; & ={\left(\lambda P^{-1}\left(n-1\right)+\hat{x}\left(n\right){\hat{x}}^{T}\left(n\right)\right)}^{-1}\\ \; & =\frac{1}{\lambda }P\left(n-1\right)-\frac{1}{\lambda }\frac{P\left(n-1\right)\hat{x}\left(n\right){\hat{x}}^{T}\left(n\right)P\left(n-1\right)}{\lambda +{\hat{x}}^{T}\left(n\right)P\left(n-1\right)\hat{x}\left(n\right)}, \end{array}$$

Define
(16)$$\begin{array}{c} G\left(n\right)\overset{▵}{=}\frac{1}{1+\left(1-\lambda \right){\hat{x}}^{T}\left(n\right)P\left(n-1\right)\hat{x}\left(n\right)}, \end{array}$$
which is the proportionate factor. Using (Equation 7) and (Equation 14) in (Equation 11), one has
(17)$$\begin{array}{cc} w(n)\; & =G\left(n\right)\left[\left(1-\lambda \right)x^{T}\left(n\right)P\left(n\right)x\left(n\right)w\left(n-1\right)+P\left(n\right)B\left(n\right)\right]\\ \; & =G\left(n\right)\left(1-\lambda \right)x^{T}\left(n\right)P\left(n\right)x\left(n\right)w\left(n-1\right)+G\left(n\right)P\left(n\right)\left(\sum_{i=0}^{n}\frac{\rho (i)}{\rho (n)}\hat{x}\left(i\right)p\left(i\right)\right)\\ \; & =G\left(n\right)\left(1-\lambda \right)x^{T}\left(n\right)P\left(n\right)x\left(n\right)w\left(n-1\right)\\ \; & \;+G\left(n\right)P\left(n\right)\left(\hat{x}\left(n\right){\hat{x}}^{T}\left(n\right)+\frac{1}{\beta (n)}P^{-1}\left(n-1\right)\right)w\left(n-1\right)\\ \; & \;+G\left(n\right)P\left(n\right)\hat{x}\left(n\right)\left(p\left(n\right)-{\hat{x}}^{T}\left(n\right)w\left(n-1\right)\right)\\ \; & =w\left(n-1\right)+G\left(n\right)P\left(n\right)\hat{x}\left(n\right)e\left(n\right). \end{array}$$

Define
(18)$$\begin{array}{c} \sigma \left(n\right)\overset{▵}{=}G\left(n\right)e\left(n\right), \end{array}$$
which can be considered as the proportionate error. The (Equation 17) is rewritten as
(19)$$\begin{array}{c} w\left(n\right)=w\left(n-1\right)+P\left(n\right)\hat{x}\left(n\right)\sigma \left(n\right). \end{array}$$

### 2.3. Momentum PFxRLS (MPFxRLS) Algorithm

It was shown in [23] that by introducing the momentum and learning rate, the performance of artificial neural network can be improved. Therefore, we extend the similar scheme for the proposed PFxRLS algorithm, and the controller update equation becomes
(20)$$\begin{array}{c} w\left(n\right)=w\left(n-1\right)+\zeta \left(n\right)P\left(n\right)\hat{x}\left(n\right)\sigma \left(n\right)+\gamma \left(n\right)\left(w(n-1)-w(n-2)\right), \end{array}$$
herein, ζ(n) and γ(n) are the learning rate and momentum factor, respectively. ζ(n) and γ(n) can be arbitrarily assigned to some values between 0 and 1, and the typical values of ζ(n) and γ(n) are close to 1 and 0, respectively. In the present study, ζ(n) and γ(n) are varied to further improve the performance of the PFxRLS algorithm.

Considering the effect of forgetting factor λ, the learning rate ζ(n) is calculated as
(21)$$\begin{array}{c} \zeta \left(n\right)=1-e^{\left(-\left(1-\lambda \right)\left|\frac{\sigma (n)}{\sigma_{v}}\right|\right)}, \end{array}$$
where $\sigma_{v}$ is the standard deviation of background noise. The momentum factor γ(n) is
(22)$$\begin{array}{c} \gamma \left(n\right)=\zeta^{2}\left(n\right)-1. \end{array}$$

### 2.4. Convergence Analysis

Define misalignment vector as
(23)$$\begin{array}{c} v\left(n\right)=w\left(n\right)-w^{*} \end{array}$$
where $w^{*}$ represents optimal controller weight. Using (Equation 20), one has
(24)$$\begin{array}{cc} v(n)\; & =\left[\zeta \left(n\right)G\left(n\right)(I-P\left(n\right)x\left(n\right)x^{T}\left(n\right))+\psi \left(n\right)I\right]v\left(n-1\right)\\ \; & \;-\gamma (n)v(n-2), \end{array}$$
where
(25)$$\begin{array}{c} \psi \left(n\right)\overset{▵}{=}1-\zeta \left(n\right)G\left(n\right)+\gamma \left(n\right). \end{array}$$

It follows from (Equation 14) that
(26)$$\begin{array}{c} I-P\left(n\right)x\left(n\right)x^{T}\left(n\right)=\lambda \frac{P^{-1}\left(n\right)}{P^{-1}\left(n-1\right)}, \end{array}$$
and from (Equation 12) that
(27)$$\begin{array}{cc} P^{-1}\left(n\right) & =\lambda P^{-1}\left(n-1\right)+\hat{x}\left(n-1\right){\hat{x}}^{T}\left(n-1\right)\\ \; & =\lambda^{n-1}\left(P^{-1}\left(0\right)+\sum_{i=0}^{n-1}\frac{1}{\lambda^{i}}\hat{x}\left(i\right)x^{T}\left(i\right)\right), \end{array}$$
such that
(28)$$\begin{array}{cc} v(n) & =(H(n)+\psi (n)I)v(n-1)-\gamma (n)v(n-2), \end{array}$$
where
(29)$$\begin{array}{cc} H(n) & \overset{▵}{=}\zeta \left(n\right)G\left(n\right)\left(I-P\left(n\right)x\left(n\right)x^{T}\left(n\right)\right)\\ \; & \overset{▵}{=}\lambda^{2}\frac{\zeta \left(n\right)G\left(n\right)\left(P^{-1}\left(0\right)+\sum_{i=0}^{n-1}\frac{1}{\lambda^{i}}\hat{x}\left(i\right)x^{T}\left(i\right)\right)}{P^{-1}\left(0\right)+\sum_{i=0}^{n-2}\frac{1}{\lambda^{i}}\hat{x}\left(i\right)x^{T}\left(i\right)}. \end{array}$$

Assuming weights are statistically independent of the input and applying expectation on both sides of (Equation 28), we have
(30)$$\begin{array}{cc} U(n) & =(Z(n)+\psi (n)I)U(n-1)-\gamma (n)U(n-2), \end{array}$$
where $U\left(n\right)=E\left\{v(n)\right\}$, $Z\left(n\right)=E\left\{H(n)\right\}$, $\psi \left(n\right)\approx E\left\{\psi (n)\right\}$ and $\gamma \left(n\right)\approx E\left\{\gamma (n)\right\}$, $E\left\{\cdot \right\}$ denotes the expectation.

Define a 2*N* dimensional state vector as
(31)$$\begin{array}{c} M\left(n\right)\overset{▵}{=}\left[\begin{array}{c} U(n)\\ U(n-1) \end{array}\right], \end{array}$$
the following recursion is obtained,
(32)$$\begin{array}{c} M\left(n\right)=\left[\begin{array}{c} Z(n)+\psi (n)I\;-\gamma (n)I\\ I\;\;\;\;0 \end{array}\right]M\left(n-1\right). \end{array}$$

The stability of the above expression is governed by the roots *r* of the determinant
(33)$$\begin{array}{c} \det\left[\begin{array}{c} Z(n)+(\psi (n)-r)I\;-\gamma (n)I\\ I\;\;\;\;\;-rI \end{array}\right]=0, \end{array}$$
for which the necessary and sufficient condition is $r_{i}<1$, i=1,2,⋯2N.

Referring to [24], we employ the following result for block matrices $\Omega_{1}$, $\Omega_{2}$, $\Omega_{3}$, and $\Omega_{4}$:(34)$$\begin{array}{c} \det\left[\begin{array}{c} \Omega_{1}\;\Omega_{2}\\ \Omega_{3}\;\Omega_{4} \end{array}\right]=\det\left[\Omega_{4}\right]\det\left[\Omega_{1}-\Omega_{2}\Omega_{4}^{-1}\Omega_{3}\right]. \end{array}$$

Assuming that $\Omega_{4}^{-1}$ exists, the following characteristic equation is derived from (Equation 33)
(35)$$\begin{array}{c} \prod_{i=1}^{N}\left(-r_{i}\right)\det\left[Z\left(n\right)+\left(\psi \left(n\right)-r\right)I-\frac{\gamma (n)}{r}I\right]=0, \end{array}$$
in which the stability is governed by the roots $r_{i}$.

Let $z_{j}\left(n\right)$, j=1,2,⋯N, is the eigenvalue of Z(n). To determine the 2N roots, we need to investigate the following quadratic form
(36)$$\begin{array}{c} {r_{j}}^{2}-\left(z_{j}\left(n\right)+\psi \left(n\right)\right)r_{j}+\gamma \left(n\right)=0. \end{array}$$

To ensure $r_{j}<1$, we need $0<z_{j}\left(n\right)+\psi \left(n\right)<1$. Using (Equation 25), we have
(37)$$\begin{array}{c} 0<\zeta^{2}\left(n\right)-\zeta \left(n\right)G\left(n\right)+z_{j}\left(n\right)<1. \end{array}$$

It follows from (27) that $z_{j}\left(n\right)\approx \lambda \zeta (n)G(n)$ if reference x(n) is a stationary signal. Using it in (Equation 37) and considering ${\left(\left(1-\lambda \right)G\left(n\right)\right)}^{2}\approx 0$, we have the following condition
(38)$$\begin{array}{c} \left(1-\lambda \right)G\left(n\right)<\zeta \left(n\right)<1+\frac{\left(1-\lambda )G(n\right)}{2}. \end{array}$$

### 2.5. Complexity Analysis

The proposed PFxRLS and MPFxRLS are summarized in Algorithm 1. We note that compared with FxRLS in [12], PFxRLS adds the computation of G(n), and MPFxRLS also adds the computation of momentum terms. The computational complexity of FxRLS, PFxRLS and MPFxRLS is shown in Table 1. When the parameters are set as N=40 and M=4, the multiplications of PFxRLS and MPFxRLS are 8212 and 8253, respectively, which are slightly more than that of FxRLS with 6568. For additions, FxRLS, PFxRLS and MPFxRLS perform similarly with 4846, 4847 and 4889, respectively. The parameter setting is consistent with the following simulations.
**Algorithm 1.** The summary of the proposed algorithms.**Initialization:****w**(0)=**0**, **P**(0)=δ**I**, δ is the initial gain, a small positive constant.**Proportionate error calculation:**$\hat{x}\left(n\right)=\sum_{m=0}^{M-1}s_{m}x\left(n-m\right)$$e\left(n\right)=p\left(n\right)-{\hat{x}}^{T}\left(n\right)w\left(n-1\right)$$P\left(n\right)=\frac{1}{\lambda }P\left(n-1\right)-\frac{1}{\lambda }\frac{P\left(n-1\right)\hat{x}\left(n\right){\hat{x}}^{T}\left(n\right)P\left(n-1\right)}{\lambda +{\hat{x}}^{T}\left(n\right)P\left(n-1\right)\hat{x}\left(n\right)}$$G\left(n\right)=\frac{1}{1+\left(1-\lambda \right){\hat{x}}^{T}\left(n\right)P\left(n-1\right)\hat{x}\left(n\right)}$$\sigma \left(n\right)\overset{▵}{=}G\left(n\right)e\left(n\right)$**Weight update (PFxRLS):**$w\left(n\right)=w\left(n-1\right)+G\left(n\right)P\left(n\right)\hat{x}\left(n\right)\sigma \left(n\right)$**Weight update (MPFxRLS):**$\zeta \left(n\right)=1-e^{\left(-\left(1-\lambda \right)\left|\frac{\sigma (n)}{\sigma_{v}}\right|\right)}$$\gamma \left(n\right)=\zeta^{2}\left(n\right)-1$$w\left(n\right)=w\left(n-1\right)+\zeta \left(n\right)P\left(n\right)\hat{x}\left(n\right)\sigma \left(n\right)+\gamma \left(n\right)\left(w(n-1)-w(n-2)\right)$

## 3. Numerical Simulations

To illustrate the effectiveness of the proposed algorithms, the following two scenarios are developed, involving different parameters for sensitivity evaluation and time-varying acoustic path and the reference signal for tracking performance testing. In the following two cases, the background noise is a white Gaussian noise with variance $\sigma_{v}=10^{-3}$. The primary path is a low-pass filter with a cutoff frequency of 0.6π and a filter length of 40. The memory length is N=40. For comparison, the mean-square error (MSE), defined as $\psi_{\mathrm{MSE}}\overset{▵}{=}10\log_{10}\left\{e^{2}\left(n\right)\right\}$, is used as reference to compare.

### 3.1. Parameters Sensibility Evaluation

In this case, the sensibility of forgetting factor λ and initial gain δ of FxRLS, PFxRLS and MPFxRLS are evaluated, respectively. The reference signal is a white Gaussian noise with variance 1. The secondary path and its estimate are set as $s=\hat{s}=\left[0,\;1,\;1.5,\;-1\right]$, which is a non-minimum phase system. Figure 2 shows the de-noising curves of FxRLS, PFxRLS and MPFxRLS with different forgetting factor λ. Figure 3 shows the de-noising curves of FxRLS, PFxRLS and MPFxRLS with different initial gain δ. Table 2 presents the attenuation results of FxRLS, PFxRLS and MPFxRLS in steady state.

It follows from Figure 2 that apart from using a large value of forgetting factor λ, PFxRLS and MPFxRLS always outperform FxRLS, and MPFxRLS performs best among them. We also note that FxRLS is affected greatly by the change of λ. As λ increases, the steady-state MSE of FxRLS is close to that of PFxRLS and MPFxRLS. However, the convergence rate is delayed greatly. Conversely, PFxRLS and MPFxRLS are all almost immune to this change. Their steady-state results vary slightly, and their convergence performance changes in a narrower range compared to FxRLS, reducing the effect of forgetting factor sensibility greatly. Therefore, it is observed that PFxRLS and MPFxRLS can provide an easier selection for forgetting factors under the conditions of faster convergence rate and lower steady-state MSE, especially the MPFxRLS.

In Figure 3, it shows that with the increase of initial gain δ, the steady-state result and convergence rate of FxRLS, PFxRLS and MPFxRLS have a slight change, as shown in Table 2. FxRLS, PFxRLS and MPFxRLS perform similarly in convergence, while PFxRLS and MPFxRLS outperform FxRLS in steady-state MSE. Moreover, we note that with the increase of δ, FxRLS has a large overshoot before convergence. When δ=100, this overshoot is 15 dB larger than that of PFxRLS and MPFxRLS. This means that PFxRLS and MPFxRLS can reduce the effect of the initial gain sensitivity compared to FxRLS.

### 3.2. Tracking Performance Testing

In this simulation, the tracking performance of the proposed algorithms is evaluated by time-varying reference signal and secondary path, respectively. The forgetting factor chooses λ=0.94 and the initial gain is δ=0.0001. In the first case, the secondary path is still the s=[0,1,1.5,−1], the reference signal is initially set as x(n)=2sin(0.2πn)+v(n) with a signal-to-noise ratio (SNR) of 20 dB, where v(n) is random noise. After the 1000th iteration number, the reference signal changes to x(n)=1.5sin(0.3πn)+v(n) with an SNR of 20 dB. The de-noising curves are shown in Figure 4a. In the second case, the reference signal is fixed with x(n)=2sin(0.2πn)+1.5sin(0.3πn)+v(n) with SNR 20 dB. Whereas the secondary path is time-varying, changing from s=[0,1,1.5,−1] to s=[0,0,1,1.2,−0.4] at the 1000th iteration number. The de-noising curves are shown in Figure 4b. The steady-state MSE of FxRLS, PFxRLS and MPFxRLS before and after convergence is shown in Table 3.

For the time-varying reference signal, we note that when the mutation occurs, FxRLS has a larger increment before convergence: almost 26 dB more than PFxRLS and MPFxRLS. At this moment, the steady-state MSE after convergence of FxRLS is 3 dB larger than PFxRLS and 6 dB larger than MPFxRLS. For the time-varying secondary path, we note that when the mutation occurs, PFxRLS and MPFxRLS converge similarly, faster than FxRLS. However, the steady-state MSE after convergence of FxRLS is 2 dB larger than PFxRLS and 5 dB larger than MPFxRLS. Therefore, it shows that PFxRLS and MPFxRLS have better tracking performance in the face of changing noise environments. Moreover, it is noted that the PFxRLS and MPFxRLS also perform better than FxRLS in steady-state MSE.

## 4. Experiment

In this part, the provided algorithms are conducted to act on an enclosed space with the approximate dimensions of 2.2 m × 1.1 m × 1.2 m, as shown in Figure 5. The noise source (a speaker) and the control output (a speaker) are located at both ends of the closed space, and the error microphone is placed in the center line between the noise source and the control output. The experiment setup is shown in Figure 6, where the real-time workshop and MATLAB/Simulink are used with a dSPACE DS1104 board. The error microphone and the two speakers are all connected to the dSPACE DS1104 board through its ADC/DAC ports via an instrumentation amplifier and low-pass filters, respectively. The instrumentation amplifier is the CROWN CE 1000 and the low-pass filter is the KROHN-hite 3550. In order to achieve global noise control in this space, the primary noise frequencies should be within the global critical frequency. Using [25], the global critical frequency of this closed space is $f_{g}\approx 269.4239$ Hz. To achieve better noise reduction performance, it is preferred to select the primary noise frequencies close to the characteristic frequencies of the closed space.

Using
(39)$$\begin{array}{c} f_{r}=\frac{c_{0}}{2}\sqrt{{\left(\frac{x}{a}\right)}^{2}+{\left(\frac{y}{b}\right)}^{2}+{\left(\frac{z}{c}\right)}^{2}} \end{array}$$
where $c_{0}$ is the sound speed, *a*, *b* and *c* are the length, width and height of the closed space, and $\left\{x,y,z\right\}\in N$. By calculating, the first eight characteristic frequencies are shown in Table 4. Consequently, the reference signal is set as
(40)$$x\left(n\right)=0.8\sin2\pi \frac{80}{f_{s}}n+0.6\sin2\pi \frac{160}{f_{s}}n+\sin2\pi \frac{240}{f_{s}}n$$
and the primary noise is given by
(41)$$p\left(n\right)=\sum_{j=0}^{L}p_{j}x\left(n-j\right)+\upsilon_{p}\left(n\right)$$
where the sampling frequency is $f_{s}$ = 1 kHz, ${\left\{p_{j}\right\}}_{j=0}^{L}$ are the impulse response coefficients of the primary path, and L=30, $\upsilon_{p}\left(n\right)$ is the background noise. The frequency response of the primary path is shown in Figure 7a, and the frequency response of the secondary path is shown in Figure 7b. The experiments are conducted using the different forgetting factor λ to show the effect of parameter sensitivity on the performance of FxRLS, PFxRLS and MPFxRLS. The normalized MSE (NMSE) result of each algorithm is used as a reference to compare.

In the first case, a small forgetting factor λ=0.96 is employed. The de-noising performance is shown in Figure 8. In the beginning, the noise source and ANC system are all off, and the microphone only measures the background noise. Then, upon starting the noise source, when the ANC system is still off, the microphone measures the mixture of background noise and primary noise. Finally, the noise source and ANC system are all on, and the microphone measures the residual error. As shown in Figure 8a, we note that the residual errors of PFxRLS and MPFxRLS are significantly smaller than that of FxRLS. Figure 8b shows the NMSE results of FxRLS, PFxRLS and MPFxRLS from ANC off to ANC on. It is observed that PFxRLS and MPFxRLS converge faster than FxRLS. FxRLS costs 3.8 s to converge, while PFxRLS needs 0.5 s and MPFxRLS needs 0.4 s. The steady-state NMSE values of FxRLS, PFxRLS and MPFxRLS are shown in Table 5. We note that the steady-state NMSE of FxRLS is about −21.2 dB, 2.6 dB greater than PFxRLS and 4.2 dB greater than MPFxRLS.

In the second case, a large forgetting factor λ=0.99 is employed. The de-noising performance is shown in Figure 9. The de-noising program is similar to the first case. It is observed that compared to case 1, the convergences of FxRLS, PFxRLS and MPFxRLS are all delayed. However, compared to FxRLS, the delays of PFxRLS and MPFxRLS are relatively small. FxRLS takes 8.6 s to converge, while PFxRLS needs 2.0 s and MPFxRLS needs 2.2 s. Furthermore, we note that in Table 5, under the condition of a larger forgetting factor, the steady-state NMSE of FxRLS, PFxRLS and MPFxRLS all become lower. However, FxRLS does not change as drastically as PFxRLS and MPFxRLS. The NMSE of FxRLS decreases by 3.3 dB, while PFxRLS decreases by 2 dB and MPFxRLS decreases by 1.8 dB.

As a result, it is observed from the above two cases that for PFxRLS and MPFxRLS, the effect of forgetting factor selection on convergence and steady-state error is smaller than FxRLS. Thence, the parameter sensitivity of PFxRLS and MPFxRLS is reduced greatly. Moreover, we note that compared to FxRLS, PFxRLS and MPFxRLS can obtain a lower steady-state error under the condition of a faster convergence rate by applying a larger forgetting factor. This advantage is very important for an adaptive algorithm acting on an ANC system.

## 5. Discussion

It is well known that for conventional FxRLS, a larger forgetting factor results in a lower steady-state error at the expense of convergence speed. Conversely, a smaller steady-state error and faster convergence rate will be obtained. The ANC using FxRLS may diverge when the forgetting factor is lower than a certain unknown boundary. The initial gain will also affect the convergence of FxRLS, but the rules, related to the noise environment, are still unclear. To overcome these problems, the difference between the posterior error and the prior error is considered in the cost function of the new algorithm. By derivation, a proportionate factor is introduced in the new algorithm. Different from any other improved FxRLS versions, the proposed new PFxRLS does not introduce any other new tuning parameters. Despite this, PFxRLS still reduces the parameter sensitivity. When the forgetting factor changes from 0.9 to 0.99, the steady-state MSE of PFxRLS only changes 4.55 dB, while conventional FxRLS changes 8.65 dB. The convergence rate of PFxRLS is delayed by almost 400 iteration numbers, while conventional FxRLS is delayed by about 650 iteration numbers.

To further improve the performance of PFxRLS, a momentum technique with a varying learning factor is introduced. To ensure stability, moreover, the convergence condition of the momentum PFxRLS (MPFxRLS) is also discussed. A clear convergence boundary of the learning factor is presented. In computational complexity, the MPFxRLS only increases the N+1 multiplication and N+2 addition compared to PFxRLS, where *N* is the memory length. However, MPFxRLS always has an obvious advantage in either convergence or steady-state error. Under the condition of a similar convergence rate, MPFxRLS can improve the attenuation level of PFxRLS by 2–3 dB, such that the tracking performance of PFxRLS is improved when facing a mutated noise environment.

Other than the extensive simulation studies including parameter sensitivity verification and tracking test, the real-time experiment conducted in a closed space is also presented. In the experiment, taking into account the global noise reduction, three characteristic frequencies with 80 Hz, 160 Hz and 240 Hz are used as the reference signal. A de-noising program, including noise resource off, ANC off and ANC on, is presented for different forgetting factors with λ=0.94 and λ=0.99. Similar results to the simulation were obtained. The convergence rate of PFxRLS and MPFxRLS is similar, 3–6 s faster than FxRLS, while the steady-state MSE of PFxRLS and MPFxRLS is 2–4 dB lower than FxRLS. Therefore, PFxRLS and MPFxRLS can obtain a lower steady-state error under the condition of a faster convergence rate.

For measurement results, the background noise is unknown, the sound field environment is complex and the measured acoustic paths do not take into account the effects of the circuit. Therefore, it is difficult to carry out the simulation and experiment in the same noise environment, resulting in incomparability between the simulation results and the measured results in terms of data. However, in terms of the de-noising trend, we compared the simulation and measurement results. Figure 2 in simulation and Figure 8 and Figure 9 in the experiment show that (1) for the same forgetting factor, PFxRLS and MPFxRLS converge faster and have a lower steady-state error than FxRLS, (2) with the increase of forgetting factor, the convergence rate and steady-state of FxRLS are affected greatly, while the convergence speed and steady-state error of PFxRLS and MPFxRLS are less affected, and (3) among FxRLS, PFxRLS and MFxRLS, the de-noising performance of MPFxRLS is best and followed by PFxRLS, while FxRLS is worst. These conclusions are consistent in the simulation and experiment.

## 6. Conclusions

In this work, a new FxRLS algorithm is proposed to overcome the effect of user-defined parameters sensitivity. By introducing the difference between the posterior error and the prior error into the cost function, a new proportionate factor is derived in the update strategy of the proposed algorithms. This proportionate factor is related to the forgetting factor and initial gain and adapts only according to the input power and user-defined parameters. In this way, it can mitigate the effects of fluctuations in these parameters. Therefore, when the forgetting factor and initial gain change in a wide range, the convergence and steady-state performance of the proposed algorithms are basically unchanged, which are close to the optimal de-noising performance. To further improve the performance of the proposed PFxRLS, a new variable momentum factor is considered in PFxRLS, and its convergence condition is also discussed. The simulation and experiment results using different forgetting factors and initial gains show that the proposed PFxRLS and MPFxRLS certainly reduce the sensitivity to these user-defined parameters. Moreover, their tracking performance and de-noising performance are also improved compared to conventional FxRLS.

## Figures and Tables

**Figure 1 sensors-22-04566-f001:**
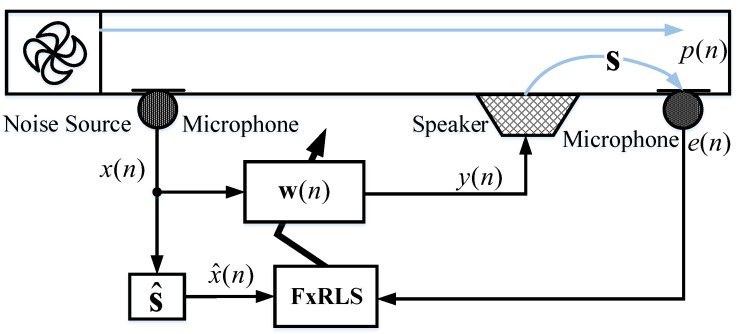
The ANC system using the FxRLS algorithm.

**Figure 2 sensors-22-04566-f002:**
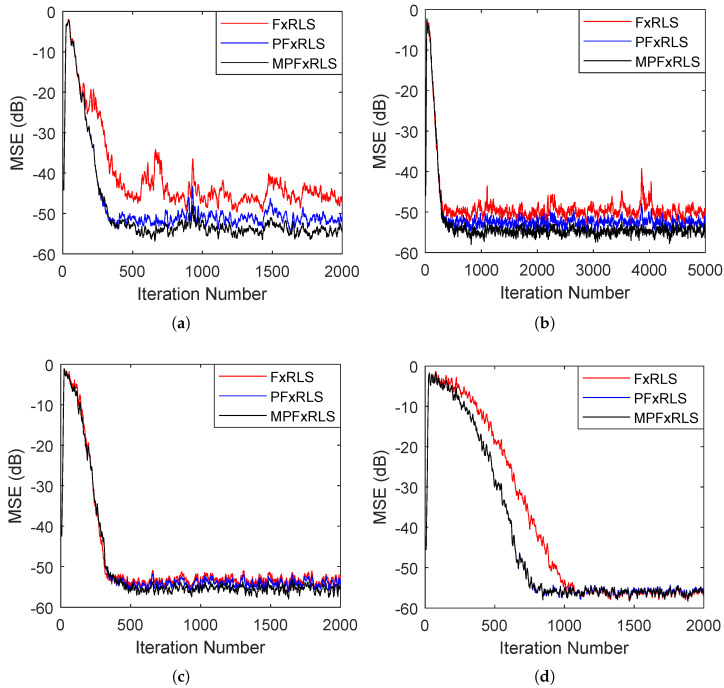
The sensibility evaluation of forgetting factor λ: (**a**) shows the MSE learning curves of λ=0.90. (**b**) shows the MSE learning curves of λ=0.93. (**c**) shows the MSE learning curves of λ=0.96. (**d**) shows the MSE learning curves of λ=0.99.

**Figure 3 sensors-22-04566-f003:**
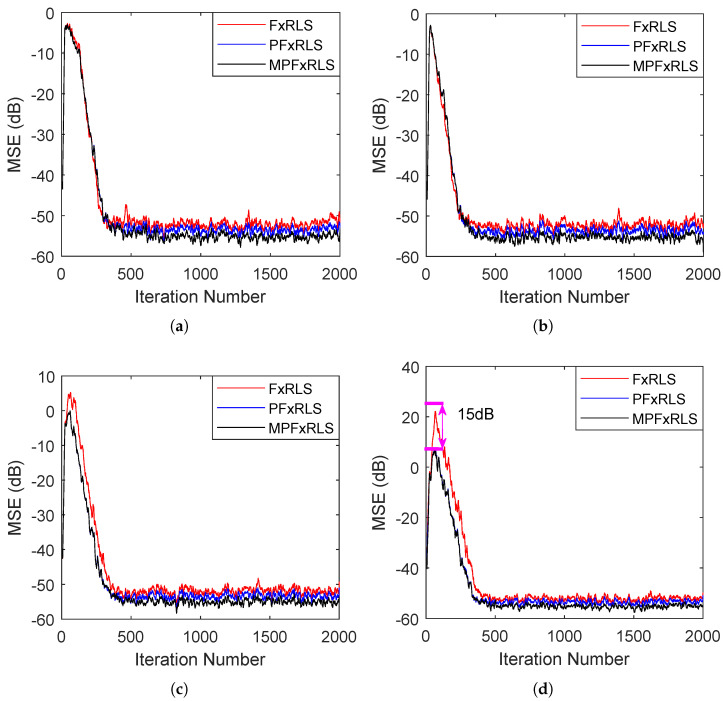
The sensibility evaluation of initial gain δ: (**a**) shows the MSE learning curves of $\delta =10^{-4}$. (**b**) shows the MSE learning curves of $\delta =10^{-2}$. (**c**) shows the MSE learning curves of δ=1. (**d**) shows the MSE learning curves of $\delta =10^{2}$.

**Figure 4 sensors-22-04566-f004:**
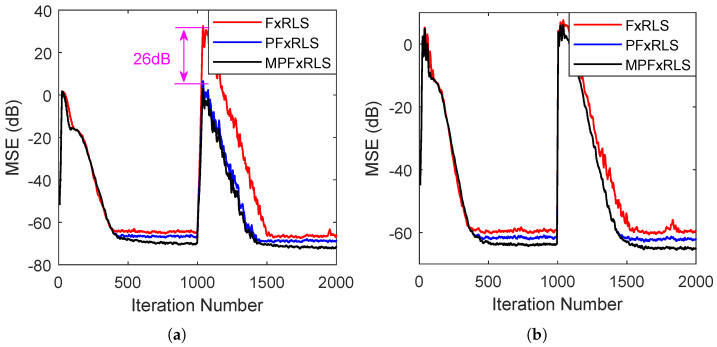
Tracking performance testing: (**a**) shows the result of changing reference signal. (**b**) shows the result of changing secondary path.

**Figure 5 sensors-22-04566-f005:**
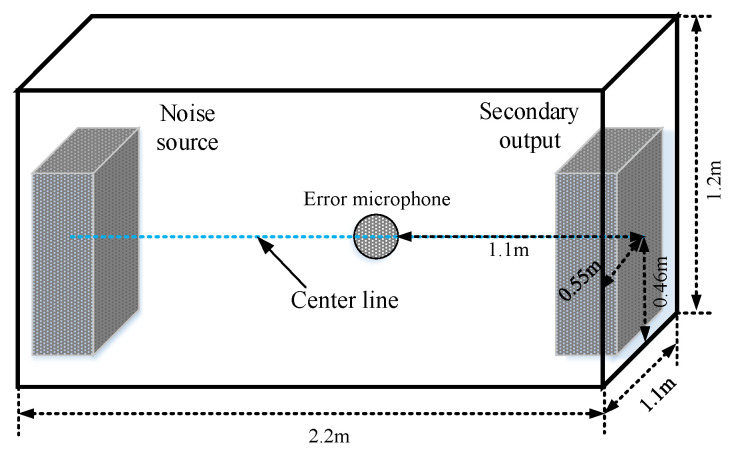
Position diagram of error microphone.

**Figure 6 sensors-22-04566-f006:**
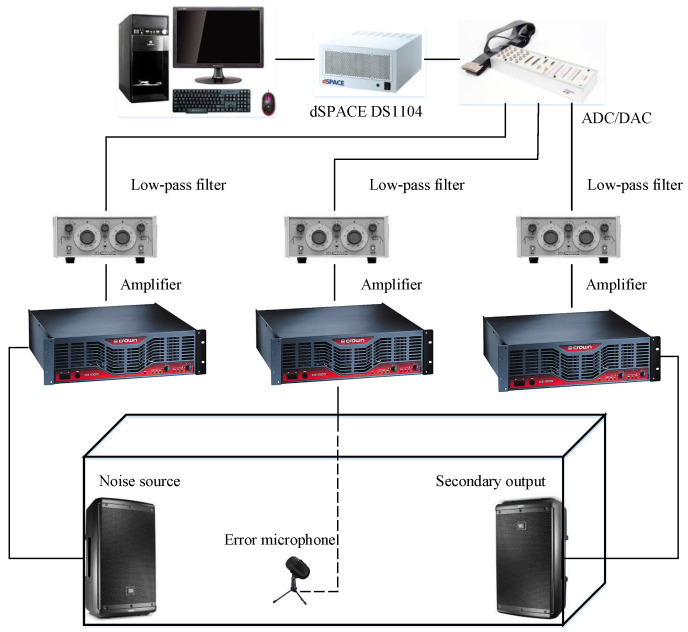
Connection diagram of experimental setup.

**Figure 7 sensors-22-04566-f007:**
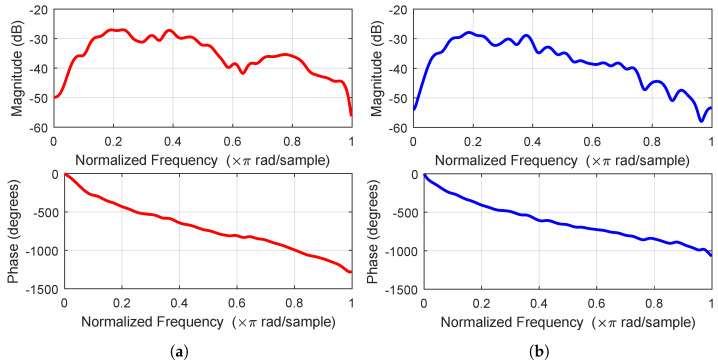
Frequency response: (**a**) shows the result of primary path. (**b**) shows the result of secondary path.

**Figure 8 sensors-22-04566-f008:**
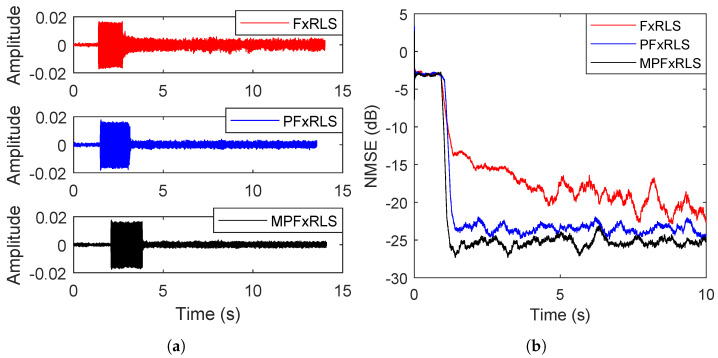
De-noising result for small forgetting factor λ=0.96: (**a**) shows the result of residual error. (**b**) shows the result of NMSE.

**Figure 9 sensors-22-04566-f009:**
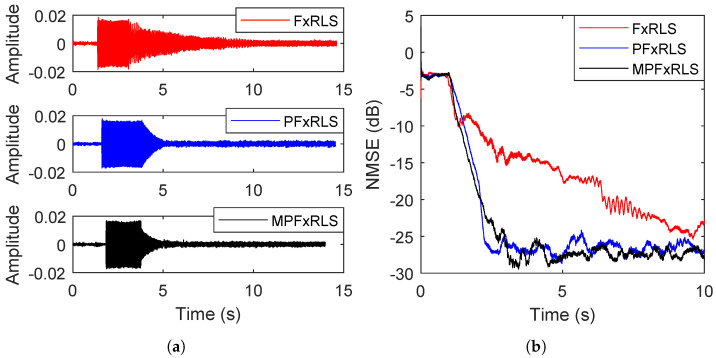
De-noising result for small forgetting factor λ=0.99: (**a**) shows the result of residual error. (**b**) shows the result of NMSE.

**Table 1 sensors-22-04566-t001:** Computational complexity comparison of FxRLS, PFxRLS and MPFxRLS.

Algorithms	×	+/−	÷
FxRLS	$4N^{2}+4N+2M$	$3N^{2}+N+2M-2$	2
PFxRLS	$5N^{2}+5N+2M+3$	$3N^{2}+N+2M-1$	3
MPFxRLS	$5N^{2}+6N+2M+4$	$3N^{2}+2N+2M+1$	4
N=40, M=4, employed in the simulation studies			
FxRLS	6568	4846	2
PFxRLS	8211	4847	3
MPFxRLS	8252	4889	4

**Table 2 sensors-22-04566-t002:** Steady-state MSE of FxRLS, PFxRLS and MPFxRLS with different parameters (dB).

λ	δ	FxRLS	PFxRLS	MPFxRLS
0.90	0.0001	−46.18	−50.36	−53.73
0.93	0.0001	−49.76	−52.84	−54.64
0.96	0.0001	−52.55	−53.81	−55.16
0.99	0.0001	−54.83	−54.91	−55.23
0.94	0.0001	−50.61	−52.97	−54.78
0.94	0.01	−51.23	−53.38	−55.21
0.94	1	−51.72	−53.13	−54.78
0.94	100	−51.35	−53.36	−54.54

**Table 3 sensors-22-04566-t003:** Steady-state MSE of FxRLS, PFxRLS and MPFxRLS for mutation noise environment (dB).

Noise Environment		FxRLS	PFxRLS	MPFxRLS
Mutated reference signal	before	−64.38	−66.58	−70.12
	after	−65.83	−68.81	−72.02
Mutated acoustic path	before	−59.62	−61.54	−63.69
	after	−59.55	−62.26	−65.15

**Table 4 sensors-22-04566-t004:** First eight characteristic frequencies of the given closed space.

Number	1	2	3	4	5	6	7	8
Frequency (Hz)	77.2727	141.6667	154.5455	161.3707	172.7871	209.6515	218.5603	223.4386

**Table 5 sensors-22-04566-t005:** Steady-state NMSE of FxRLS, PFxRLS and MPFxRS in the real-time experiment (dB).

Cases	FxRLS	PFxRLS	MPFxRLS
λ=0.94	−21.2	−23.8	−25.4
λ=0.99	−24.5	−25.8	−27.2

## Data Availability

Not applicable.

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
