# Peer review of "A New Proportionate Filtered-x RLS Algorithm for Active Noise Control System"

_sensors, 2022, doi:10.3390/s22124566_

Round 1
Reviewer 1 Report
Comments and Suggestions for Authors
„ A New Proportionate Filtered-x RLS Algorithm for Active Noise Control System” is an interesting paper.
Paper needs to be further improved in order to be recommended for publication. The paper needs to be carefully revised to improve terminology and readability.
Comment 1: The authors use too long sentences throughout the manuscript. After reaching the end of the sentences, the reader cannot remember how the sentence started. I recommend shortening at least some sentences that are too long.
Comment 2: The whole paper needs English corrections.
Comment 3: The reviewer thinks that the authors could add more information about the possibilities of improving the noise reduction performance of the ANC system and corresponding adaptive algorithms.
Comment 4: Title Tab. 5 First eight characteristic frequencies of the given closed space (without unit) but in Table 5, unit of frequency must be added.
Comment 5: There should be a space between the number and the unit (lines – 153, 156, 160, 161 etc).
Comment 6: Discussion missing. The authors should provide discussion and compare their results with the results of other authors.
Comment 7: The reviewer recommends revising the conclusions.
Comment 8: Is it possible to use both factors in the simulation at the same time?
Comments 9: There is no comparison of simulation results with measurement results. I recommend adding.
The research presented in the article is interesting. Paper needs to be further improved in order
Reviewer 2 Report
This paper proposes a new proportionate FxRLS (PFxRLS) algorithm, without introducing new turning parameters, which not only reduces the sensitivity of the forgetting factor and initial gain but also improves the denoising level and tracking performance of conventional FxRLS.
Advantages of the paper:
- This manuscript is well written and has good significance.
- Novel, original FxRLS (PFxRLS) algorithm is suggested in this study.
- Good picture formatting with readable fine text and structures.
- Validation of the algorithm is conducted via experiments and simulations.
Note/Recommendations:
The whole paper should be reviewed carefully, to correct all the typing/spelling errors.
Conclusion:
As a referee, I could not find any defect in this manuscript. I'd suggest accepting the paper in its present form.
Round 2
Reviewer 1 Report
The authors have made detailed adjustments to the original text. After editing, shortening some long sentences, the text of the article is more acceptable, the orientation in the text has improved. The authors have added all the required information to the text.
The revised document has reached an acceptable level for publication.